# Evaluation of a Four-Week Online Resilience Training Program for Multiple Sclerosis Patients

**DOI:** 10.3390/healthcare12060620

**Published:** 2024-03-09

**Authors:** Lotte Bock, Madiha Rana, Lara Westemeyer, Majeed Rana

**Affiliations:** 1Institute of Psychology in Education, Leuphana Universität Lüneburg, Universitätsallee 1, DE-21335 Lüneburg, Germany; 2Department of Psychology, Europäische Fernhochschule Hamburg, University of Applied Sciences, Doberaner Weg 20, DE-22143 Hamburg, Germany; madiha.rana@euro-fh.de; 3Bundeswehrkrankenhaus Hamburg, Klinik Für Neurologie, Lesserstraße 180, DE-22049 Hamburg, Germany; lara.westemeyer@web.de; 4Department of Oral and Maxillofacial Surgery, Heinrich Heine University Duesseldorf, Moorenstrasse 5, DE-40225 Duesseldorf, Germany; rana@med.uni-duesseldorf.de

**Keywords:** resilience, stress, online training, meditation, mindfulness

## Abstract

The diagnosis of a chronic disease, such as multiple sclerosis, has both psychological and physical effects. Living with the disease and its uncertain consequences requires a great deal of psychological resilience in order to employ more comprehensive coping strategies in stressful situations. This study investigated the effect of a four-week online self-directed resilience training program on the perception of psychological resilience among multiple sclerosis patients. A total of 94 MS patients were recruited for a randomised controlled trial. The experimental group underwent a 28-day online self-directed training program consisting of daily exercises aimed at strengthening a resilient mindset. Psychological resilience was measured through self-assessment immediately before, immediately after, and three months after the training. A repeated measures ANOVA revealed a statistically significant improvement in the perception of four factors related to stress: perceived worries, tension, joy, and demands. Two resilience coping strategies were measured, of which one, a resilient orientation, improved significantly in the short and long term. The study suggests that online self-directed resilience training might provide an easily accessible, low-cost option for patients with MS to improve their psychological resilience. This is a pilot study to assess the general applicability to people with MS. Future studies should examine the transferability of results in relation to disease stage and co-morbidities.

## 1. Introduction

Being diagnosed with an incurable chronic disease such as multiple sclerosis (MS) can significantly alter an individual’s life, resulting in physical limitations and profound psychological effects. As the prevalence of MS increases worldwide [1], it is becoming increasingly important to identify and implement accessible ways to enhance patients’ resilience so that they can effectively cope with the challenges of the disease and strengthen their ability to overcome it. 

Resilience refers to an individual’s ability to adapt and withstand life’s challenges. It has become increasingly important, particularly during the COVID-19 pandemic [2]. Improving psychological resilience in patients with multiple sclerosis may be a promising approach to enhancing their quality of life.

Previous studies have suggested that resilience may be a stable personality trait. However, it is now understood that resilience is dynamic and focused on individual development [3] rather than a fixed perspective. Foundational studies [4,5,6] have collectively influenced the perception of resilience to that of being malleable and flexible, challenging the original notion that it is a fixed personality trait. As resilience is not a stable trait, it can be improved through training. The current research literature indicates that such interventions are promising.

A pilot evaluation was conducted to assess the effectiveness and feasibility of a group resilience training for individuals with MS. The training was based on acceptance and commitment therapy (ACT) and showed promising results [7]. Furthermore, a study examined the relationship between psychological resilience and social and occupational performance in people with MS, highlighting the importance of resilience in this population [8]. Arab et al. [9] conducted a study on the effects of a distancing program on the sense of coherence in MS patients. Their study highlights the relevance of resilience-promoting interventions in this population.

Additionally, Ploughman et al. [10] investigated the impact of resilience on healthy aging in MS, suggesting the potential role of resilience in improving the overall well-being of people with MS. Broche-Pérez et al. [11] investigated the mediating role of psychological resilience in the relationship between fear of relapse and quality of life in individuals with MS. Their study highlights the significance of resilience in managing disease-related concerns. Furthermore, a study discovered that psychological resilience played a mediating role in the correlation between perceived neuropsychological impairment and quality of life in individuals with MS. This implies that interventions aimed at building resilience may have a positive impact on the psychological well-being and quality of life of those with MS [12].

In summary, current findings highlight the need for more research on interventions that can reduce stress and increase resilience by people with MS [13,14]. A meta-study suggests that interventions aimed at enhancing resilience and stress management have the potential to positively impact the well-being and quality of life of people with MS, but more research is needed [15].

This pilot study aimed to investigate the effects of online self-directed training on enhancing personal resilience and reducing stress perception in individuals with MS. Previous research has demonstrated the potential of self-directed online training to prevent or reduce stress, both generally and specifically in the context of online mindfulness training [16,17,18,19,20,21]. The existing research suggests potential benefits and highlights the need for further investigation [22].

## 2. Materials and Methods

The Euro-FH Ethics Committee approved this study (EKEFH04/23), and all participants provided informed written consent. This study is a randomised controlled trial with three assessment points, baseline (T1), post-intervention (T2), and long-term (T3), conducted between March and July 2023.

A total sample size of *n* = 44 was determined using an ANOVA repeated measure analysis (G*Power 3.1) with an effect size of d = 0.25, a-error = 0.05, and b-error = 0.95. The study recruited 94 people with MS from the Department of Neurology, Bundeswehrkrankenhaus Hamburg, and the German Multiple Sclerosis Society (DMSG). Patients at the Department of Neurology at Bundeswehrkrankenhaus Hamburg were personally contacted and informed about the study’s procedure and purpose. Consent for data use was obtained from the internal data protection officer of the hospital. Additionally, the project was communicated to the regional associations of the DMSG via email, with a clear and transparent explanation of the aim and content of the study. The DMSG distributed the information to its members and advertised the study to those who were interested. Interested participants contacted the research team via mail. A total of 137 individuals expressed interest in the study, with 56 from the Bundeswehrkrankenhaus Hamburg and 81 from DMSG. The study’s inclusion criteria required participants to be legal adults over 18 years old and have a clinical diagnosis of multiple sclerosis, a chronic neurological condition, evaluated by medical professionals. This ensured a homogeneous participant group, specifically focusing on those affected by the condition under investigation. Additionally, participants were required to demonstrate a willingness to allocate a designated amount of time each day (20 min) towards engaging in the self-directed resilience training program. This commitment reflects the importance of adherence to the intervention, which is crucial for evaluating its efficacy and effectiveness. Participants were required to complete three questionnaires during the testing period to provide essential data for the research. This study excluded individuals who did not meet the specified inclusion criteria. Ninety-four individuals met the inclusion criteria and completed the final anonymous registration, indicating successful recruitment of participants who met the study’s requirements.

The research group randomly assigned 94 participants to either the experimental group (EG) (*n* = 47) or the waitlist control group (CG) (*n* = 47) using an online randomisation tool. All participants were informed of the study design and knew that they would either be assigned to the EG and start the program first, or to the CG and start the program four weeks later. Participants were then asked to complete the first questionnaire (T1) and were then informed of their allocation. After completing the questionnaire, the EG began the online program. The CG was informed that they would gain access to the program four weeks later, after completing the second questionnaire (T2). All 94 participants completed the T2 questionnaire after the EG finished the program. To measure the long-term effect of the program on the experimental group (EG), participants were required to complete a questionnaire for the third time (T3) three months after the program had ended. Participants were informed that only complete data sets consisting of three filled-out questionnaires would be considered for the research.

Despite three reminder emails, thirty participants did not complete all three questionnaires and were excluded from the study. Due to strict privacy policies and anonymity, it was not possible for the research team to contact these participants to find out why they did not complete the third questionnaire. This study analysed the long-term effect of the program after three months on the 63 participants who completed all three questionnaires (T1, T2, and T3). Twenty-nine of the remaining participants belonged to the CG and thirty-four to the EG. The CG was granted access to the program immediately after the EG had completed it and after they had filled out the T2 questionnaire. Out of the original 47 CT participants, 29 chose to participate in the program.

The self-developed program was based on established stress theories, such as Lazarus and Folkman [23], and current research on resilience and the effects of online training [24,25,26,27,28]. The participants in the experimental group (EG) received two emails per day, one in the morning and one in the evening, for 28 consecutive days. They were free to integrate the videos and inputs from the resilience training program into their daily routine at their convenience. The morning sessions focused on improving resilience, while the evening sessions were used for reflecting on the daily content and exercises. This was done using a journal, a method that involves writing down thoughts. Participants were free to use their mobile phones or individual pieces of paper to make notes in a diary-like format. The tasks for the evening included prompts such as ‘Write down three to five things that you were grateful for (or very satisfied with) today’ and ‘Identify the people you feel connected to, the places where you feel a sense of belonging, and the larger communities you are a part of’. The morning training content was sent to program participants via email every day at 7am. Each morning, the training session consisted of a video-based ‘learning nugget’ on a resilience-related topic, followed by an activity that reinforced the newly acquired knowledge throughout the day. The session concluded with a relaxation exercise, which could be a breathing exercise, meditation, or an easy-to-practice yoga exercise. The total duration of the daily sessions was approximately 20 min, with 15 min in the morning and 5 min in the evening. Before the program began, participants received a brief introductory and welcome video explaining how to organise the exercises according to their preferences. This provided flexibility in watching the videos, as they did not need to be viewed all at once.

During the first week of the program, the focus was on understanding resilience and establishing a daily routine of reflection and relaxation. The content covered neuropsychological principles, as well as topics such as sleep, activating personal resources, rest and relaxation, stress, and emotional regulation. One of the key takeaways for participants was to pay attention to small things that can improve their day and to write them down. The participants of the program were given instructions to practice different breathing and concentration exercises [23,29,30]. In the second week, the focus shifted towards solution orientation. This involved addressing the transactional stress model and cognitive distortions, as well as exploring acceptance and control, and other coping and solution strategies. An exercise during the week involved questioning one’s own perceptions and practicing saying ‘yes’. In addition, the program incorporated short meditation and yoga exercises for relaxation [31,32]. In the third week, the program focused on reframing as a coping strategy to support resilience. Participants learned to counter irrational thoughts and re-evaluate their feelings and thought processes. This week covered the topic of neuroplasticity and attribution styles. Participants were encouraged to identify their thoughts and recognise irrational thoughts or self-regulation. The relaxation exercises aimed to promote calmness and relaxation while observing one’s own thoughts [33,34]. The last week’s focus was on relationships and their effects, covering topics such as connection, empathy, compassion, and dealing with difficult people, as relationships are considered a fundamental psychological need. The ‘learning nugget’ also addressed self-efficacy, resource management, and self-criticism. The participants were encouraged to demonstrate their understanding and compassion for others. Furthermore, the topic of gratitude was explored as a means of strengthening one’s own resources [35,36]. The training program aimed to improve individuals’ perception of stress and resilience through daily engagement with tailored exercises and content.

Participants evaluated their stress perception and resilience using a combined questionnaire comprising two assessments: the Resistance Orientation–Regeneration Orientation Scale (Re-Re Scale), developed by Otto and Linden [37], and the Perceived Stress Questionnaire, developed by Fliege et al. [38].

The Resistance Orientation–Regeneration Orientation Scale (Re-Re Scale) developed by Otto and Linden [37] is utilised to document stress-related procedures. The Re-Re Scale comprises 20 items, categorised into two subscales. The ‘Resistance Orientation’ subscale comprises 10 items that evaluate resilience and individual behaviour towards achieving goals. The scale includes examples such as ‘When striving for a goal, personal emotions should not be a factor’ and ‘External factors do not impress me’. The ‘Regeneration Orientation’ subscale consists of 10 items that measure the inclination towards self-care. Examples of these items include ‘During stressful periods, recovery time is especially important’ and ‘I focus on my positive attributes when looking in the mirror’. The program participants categorised their answers on a 5-point Likert scale ranging from ‘1 = strongly disagree’ to ‘5 = strongly agree’.

The reliability of the two subscales was indicated using Cronbach’s alpha. The ‘Resistance Orientation’ scale resulted in α = 0.93, while the ‘Regeneration Orientation’ scale resulted in α = 0.92, indicating excellent internal consistency. The Re-Re scale was analysed by calculating the mean values of the respective subscales.

The Perceived Stress Questionnaire (PSQ) developed by Fliege et al. [38] is used to assess an individual’s subjective perception of stress. For this study, we used the German-language short version of the questionnaire, which consists of 20 items divided into four subscales: ‘Worry’, ‘Tension’, ‘Joy’, and ‘Demands’. These subscales aim to demonstrate how stressful stimuli are perceived, evaluated, and processed. The subscale ‘Joy’ is to be understood as ‘lack of joy’. The PSQ includes items such as ‘You have the feeling that too many demands are being placed on you’, ‘You are full of energy’, and ‘Your problems seem to be piling up’. These items are categorised on a 4-point Likert scale ranging from ‘almost never’ to ‘most of the time’. In addition to the four individual scales, the total score of the PSQ can be calculated from all items.

The internal consistency of the PSQ was assessed using Cronbach’s alpha. The PSQ total score had a value of α = 0.86, while the four subscales had values ranging from α = 0.8 to α = 0.85, indicating good reliability in each case. Similarly high values were obtained for split-half reliability. To calculate the individual scales, the respective item values were added up according to the evaluation manual. The assessments T1, T2, and T3 were conducted through an online questionnaire tool called ScoSci Survey.

A mixed factorial repeated measures analysis of variance (ANOVA) was conducted to examine differences over time. No unusual or abnormal data were detected. Violations of sphericity were addressed using the Greenhouse–Geisser correction for values less than 0.75 or the Huynh–Feldt correction for values greater than 0.75 [39]. Levene’s test was used to assess the assumption of homogeneity of variance. If the assumption of homogeneity of variance was met, we performed post hoc multiple comparison tests using Tukey’s approach [40]. If homogeneity was not found, we used Holm’s method for post hoc tests [41]. We set the significance level for the mixed factorial repeated measures ANOVA and its associated post hoc tests at 0.05, based on our a priori power analysis.

## 3. Results

This study involved an experimental group of 34 participants, with 74% of them being female (*n* = 25) and 26% male (*n* = 9 (Figure 1 and Figure 2). The mean age of this group was 49.1 years (SD = 11.139), ranging from 27 to 65 years. The waitlist control group (*n* = 29) comprised 62% females (*n* = 18) and 38% males (*n* = 11) with a mean age of 49.31 (SD = 9.111) ranging from 30 to 67 years (Table 1).

Both scales exhibited a normal distribution (*p* > 0.05), except for the control group T1 for the PSQ. Assuming homogeneity of variance on all scales, a repeated measures ANOVA with Huynh–Feldt correction revealed a significant interaction between time and group in relation to the PSQ score (F(1.854, 113.076) = 13.880, *p* < 0.001, η^2^p = 0.185). Please refer to Figure 3 for more details.

A post hoc analysis using Bonferroni correction supported implicit differences between the experimental and waitlist control groups over time. The intervention led to a significant improvement in scores (t(34) = −6.702, *p* < 0.001, MD = −18.822 points, 95% CI [−27.231, −10.413], d = 0.984) from T1 to T2, indicating a substantial effect. Additionally, there was a significant difference between T1 and T3, with a mean difference of −13.235 points, 95% CI [−21.644, −4.826], d = 0.692, t(34) = −4.712, *p* < 0.001.

However, there was no discernible difference between T1 (control) and T2 (control). After the waitlist control group completed the intervention, their scores showed a notable increase. The mean difference (MD) from T2 (control) to T3 (control) was −12.071 points, with a 95% confidence interval of −21.176 to −2.966, and d = −0.631. This resulted in a significant improvement in scores, t(29) = −3.969, *p* = 0.002, indicating a substantial effect (see Table 2).

### 3.1. PSQ Worries Subscale

A repeated measures analysis of variance (ANOVA) was conducted to investigate the relationship between time and group on the PSQ score for the Worries subscale, with a Huynh–Feldt correction applied. The results indicated a statistically significant interaction between time and group on the PSQ score (F(1.729, 241.168) = 13.880, *p* = 0.003, η^2^p = 0.099), as shown in Figure 4.

Post hoc tests with Bonferroni correction revealed significant differences over time between the experimental and waitlist control groups. The intervention led to a significant increase in scores, with a mean difference of −17.843 points (95% CI [−28.328, −7.358], d = 0.740, t(34) = −5.095, *p* < 0.001) from T1 to T2. This effect size is considered large. Additionally, there was a significant difference between T1 and T3, with a mean difference of −11.764 points (95% CI [−22.249, −1.279], d = −0.488, t(34) = −3.359, *p* = 0.013).

No significant difference was observed between T1 (control) and T2 (control), or between T2 (control) and T3 (control).

### 3.2. PSQ Tension Subscale

A repeated measures analysis of variance (ANOVA) was conducted on the PSQ Tension subscale, with a Huynh–Feldt correction. The results showed a significant interaction between time and group on PSQ score [F(1.868, 113.924) = 12.774, *p* = 0.001, η^2^p = 0.173], as illustrated in Figure 5.

Post hoc tests with Bonferroni correction revealed significant differences between the experimental and waitlist control groups over time. The intervention led to a significant increase in scores. The mean difference was −21.176 points (95% CI [−30.784, −11.567], d = −1.011) from T1 (test) to T2 (test), indicating a large effect size (t(34) = −6.599, *p* < 0.001). Additionally, there was a significant difference between T1 (test) and T3 (test), with a mean difference of −16.862 points (95% CI [−26.470, −7.254], d = −0.805, t(34) = −5.254, *p* < 0.001).

No significant difference was found between T1 (control) and T2 (control). However, a significant difference was found between T2 (control) and T3 (control) (t(29) = −4.102, *p* = 0.001). The mean difference (MD) was −14.254 points with a 95% confidence interval (CI) of [−24.658, −3.850], indicating a large effect size (d = −0.681).

### 3.3. PSQ Joy Subscale

A repeated measures analysis of variance (ANOVA) was conducted on the Joy subscale of the Perceived Stress Questionnaire (PSQ) with a Huynh–Feldt correction. The results revealed a significant interaction effect between time and group (F(1.996, 121,774) = 13,798, *p* < 0.001, η^2^p = 0.184), as shown in Figure 6.

Subsequent post hoc tests with Bonferroni correction confirmed time-related differences between the experimental and waitlist control groups. The intervention resulted in a significant improvement in scores (MD = −19.411 points, 95% CI [−28.939, −9.884], d = −0.920, t(34) = −6.100, *p* < 0.001), indicating a large effect size. Furthermore, there was a substantial difference between T1 (test) and T3 (test) (MD = −14.118 points, 95% CI [−23.645, −4.590], d = −0.669, t(34) = −4.436, *p* < 0.001).

There was no significant difference found between T1 (control) and T2 (control). However, T2 (control) showed a significant difference from T3 (control) (t(29) = −4.804, *p* < 0.001, MD = −16.552 points, 95% CI [−26.869, −6.236], d = −0.784), representing a large effect size.

### 3.4. PSQ Demands Subscale

A repeated measures analysis of variance (ANOVA) with a Huynh–Feldt correction was conducted on the PSQ Demands subscale. The results revealed a statistically significant interaction between time and group on the PSQ score [F(1.910,116,434) = 6.58, *p* = 0.002, η^2^p = 0.100], as shown in Figure 7.

Bonferroni post hoc tests indicated significant differences between the experimental and waitlist control groups over time. The intervention led to a significant improvement in scores (mean difference = −16.864 points, 95% CI [−28.027, −5.700], d = −0.719, t(34) = −4.523, *p* < 0.001) from T1 (test) to T2 (test). However, there were no significant differences between T1 (test) and T3 (test), or between T2 (control) and T3 (control).

### 3.5. Resistance Orientation–Regeneration Orientation (Re-Re) Scale

The scale comprises two subscales: Resistance and Regeneration. A repeated measures ANOVA for the Re-Re scale as a whole did not reveal any statistically significant interaction between time and group. However, the Regeneration Orientation subscale demonstrated significant improvement, whereas the Resistance–Regeneration subscale did not.

A repeated measures ANOVA with a Huynh–Feldt correction revealed a significant interaction between time and group on the PSQ score for the subscale of Worries (F(1.817, 110.845) = 10.094, *p* = 0.001, η^2^p = 0.142). The results are presented in Figure 8.

Furthermore, post hoc tests using the Bonferroni correction confirmed significant differences between the experimental test group and the waitlist control group over time. The intervention led to a significant increase in test scores, with a large effect size (d = −0.789), t(34) = −5.845, *p* < 0.001, and a mean difference (MD) of −0.544 points, 95% CI [−0.823, −0.265], between T1 (test) and T2 (test). Additionally, a significant difference was observed between T1 (test) and T3 (test), with a mean difference of −0.479 points, 95% CI [−0.758, −0.201], d = −0.695, t(34) = −5.150, *p* < 0.001.

No significant difference was found between the T1 and T2 control groups. However, the control group T3 exhibited a significant difference compared to T2. The mean difference (MD) was −0.441 points, with a 95% confidence interval (CI) of [−0.743, −0.140]. This resulted in a t-value of −4.379 and a p-value of less than 0.001. The effect size was considered medium, with d = −0.640.

## 4. Discussion

In summary, this study shows a consistent improvement in various aspects of psychological well-being in the experimental group. The intervention had an immediate positive effect, as evidenced by significant increases in all scales of the Psychosocial Well-being Questionnaire (PSQ) from T1 (baseline) to T2 (immediately after the training program). The effectiveness of the program in positively influencing the psychological state of the participants was immediately apparent.

Additionally, this study shows sustained long-term improvements (T3) after three months, particularly in the PSQ scales measuring worries, joy, and tension, indicating that the program’s benefits persisted in these crucial dimensions of psychosocial health. Although the Demands subscale did not show significant effects, positive trends in other PSQ subscales suggest an overall positive impact on various aspects of psychological well-being. It is worth noting that the experimental group showed significant improvements in the Regeneration subscale, both in the short term (T2) and the long term (T3) on the Resilience–Recovery (Re-Re) scale. This highlights the effectiveness of the intervention in facilitating psychological recovery among participants. However, measures related to resilience did not show a significant improvement, revealing nuanced effects across different domains of psychological well-being within the experimental group.

A decline in the program’s effectiveness is observed between T2 and T3, which is consistent with the typical pattern seen in the post-trial period of interventions [42]. However, the study emphasises the enduring significance of the program by highlighting that the improvements observed from T1 to T3 remain statistically significant across most scales. This suggests that the positive impact of the program persists over the long term, even if its immediate effect diminishes.

The control group (CG), which was granted access to the program after the main trial period, demonstrates significant improvement between T2 and T3. It is important to note that enrolment in the program was voluntary for the CG, and two-thirds of the participants chose to engage in the program. The improvement in the CG, measured two months after the conclusion of the course (T3), strongly indicates the effectiveness of the program. This supports the notion that the intervention had a meaningful and lasting impact on the well-being of participants.

This study provides evidence indicating the potential of self-directed online programs in bolstering resilience and diminishing perceived stress levels among individuals diagnosed with multiple sclerosis (MS). These findings offer a glimpse into the efficacy of such interventions in addressing the challenges faced by individuals coping with chronic health conditions.

The findings of this study contribute significantly to the growing body of evidence highlighting the vital role of resilience in individuals coping with multiple sclerosis (MS) [1]. Consistent with previous research and the existing literature, this study emphasises the potential effectiveness of resilience-focused, stress-reducing interventions in addressing the unique challenges faced by individuals dealing with MS in general.

Future studies should consider disease severity, neurological disability, and co-morbidities in people with MS, as these factors may significantly influence response to resilience interventions and overall experience of the disease. Investigating the interaction of variables with the effectiveness of self-directed online programs may help to identify subgroups of patients who may benefit most. In addition, investigating the relationship between age and gender and the effectiveness of resilience interventions may provide valuable insights into the particular challenges faced by different demographic groups within the MS population, supporting the development of targeted interventions tailored to specific needs.

The promising results of this preliminary study suggest that self-directed online resilience training has the potential to significantly reduce stress in people with MS. Future research should focus on investigating the long-term effects of such interventions, particularly over longer periods of six to twelve months post-intervention. Strategies to improve participant retention and engagement are crucial to minimise dropout rates in longitudinal studies. The inclusion of a third control group, receiving a placebo intervention or an alternative stress reduction program, may help to elucidate the true effects of the intervention beyond the participants’ perceptions.

In conclusion, immediate post-diagnosis support, particularly through resilience courses, is recommended as a crucial aspect of coping with the psychological impact of MS. Resilience-building interventions can reduce the perception of stress and argue for proactive measures to provide psychological support. The suggestion that the course material should be integrated into daily life for long-term use is supported by the positive results observed three months after the intervention, suggesting lasting benefits. Widespread use of self-directed online interventions could have far-reaching benefits for people with MS and contribute to their overall well-being. This study provides important insights for further research and comprehensive care strategies to improve the mental well-being of people facing the challenges of multiple sclerosis.

## 5. Conclusions

The study sheds light on how targeted interventions can support the holistic well-being of people with multiple sclerosis (MS), building on and extending existing research. While our findings provide initial support for the efficacy of self-directed online programs for this population, future research should take a more comprehensive approach, controlling for various covariates and conducting subgroup analyses to deepen our understanding and optimise the impact of such interventions.

These findings underscore the importance of exploring alternative methods of supporting people with MS, particularly in alleviating the psychological and emotional burden associated with the disease. Our study suggests that engaging participants in self-directed online programs is a practical and accessible way to promote resilience and improve overall well-being in this population.

While these preliminary findings are promising, they also serve as a catalyst for further research into self-directed online interventions for chronic disease management. It is essential that these findings inspire ongoing research aimed at elucidating the mechanisms underlying the observed benefits and refining the design and delivery of such programs.

Furthermore, the implications of these findings extend beyond MS and can be extrapolated to a wider range of chronic health conditions. This underscores the potential of self-directed online interventions to reduce stress, increase life satisfaction, and build resilience in individuals facing a range of medical challenges. As such, these initial findings serve as a clarion call for sustained interdisciplinary collaboration and innovation in the pursuit of more effective and accessible approaches to improving the quality of life of people with chronic conditions.

## Figures and Tables

**Figure 1 healthcare-12-00620-f001:**
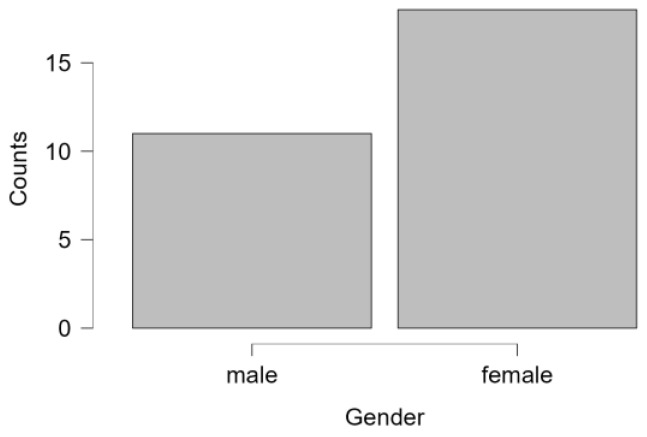
Sex distribution control group.

**Figure 2 healthcare-12-00620-f002:**
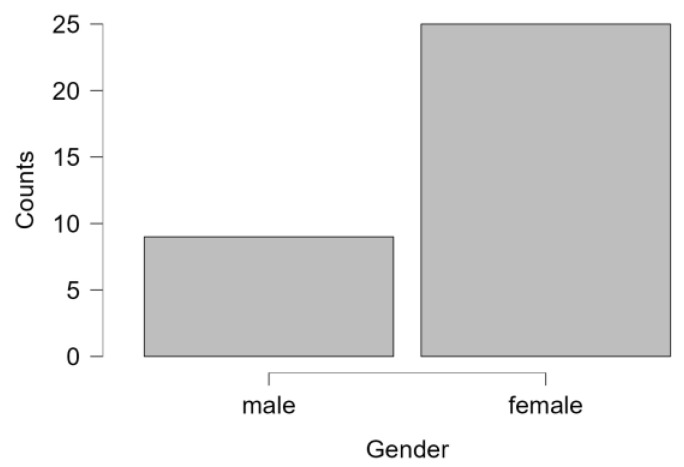
Sex distribution experimental group.

**Figure 3 healthcare-12-00620-f003:**
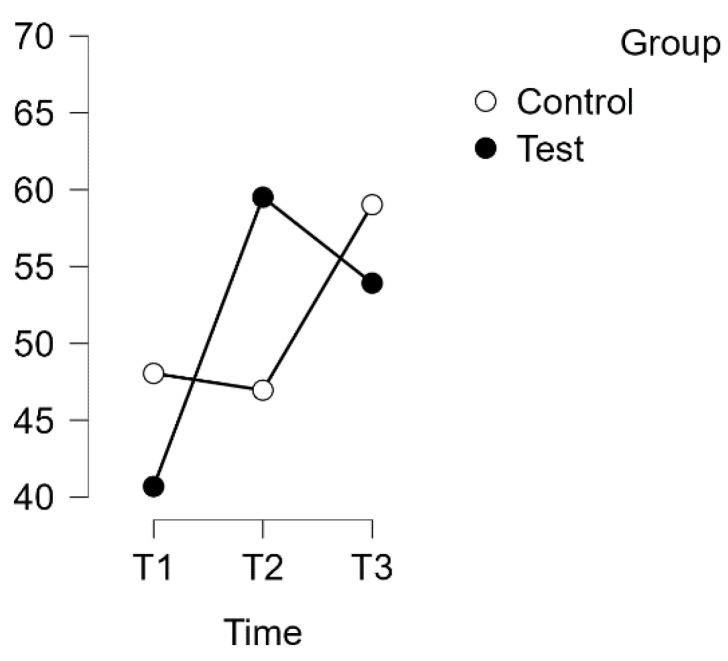
Estimated marginal means for PSQ complete score.

**Figure 4 healthcare-12-00620-f004:**
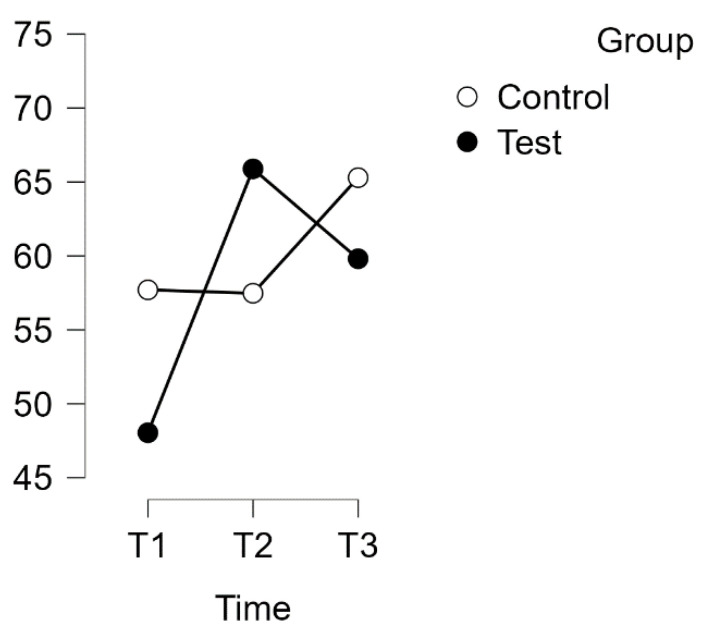
Estimated marginal means for PSQ subscale: Worries.

**Figure 5 healthcare-12-00620-f005:**
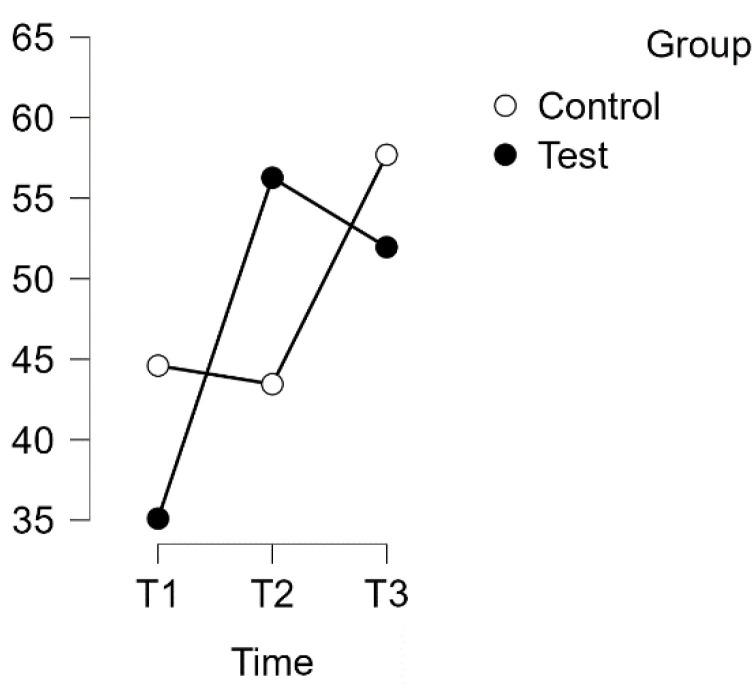
Estimated marginal means for PSQ subscale: Tension.

**Figure 6 healthcare-12-00620-f006:**
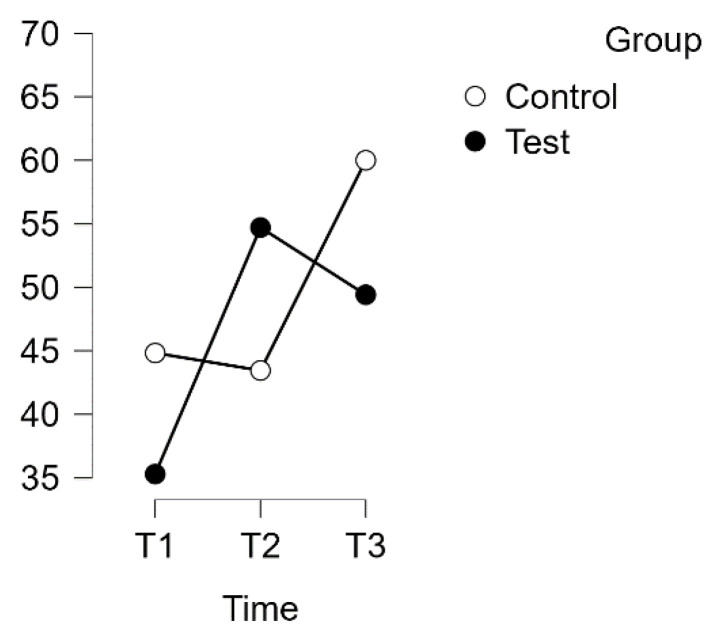
Estimated marginal means for PSQ subscale: Joy.

**Figure 7 healthcare-12-00620-f007:**
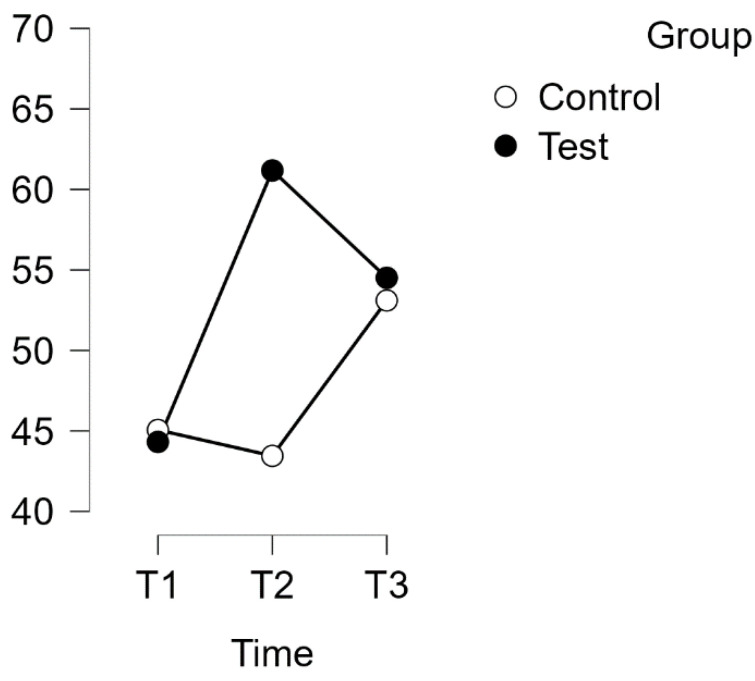
Estimated marginal means for PSQ subscale: Demands.

**Figure 8 healthcare-12-00620-f008:**
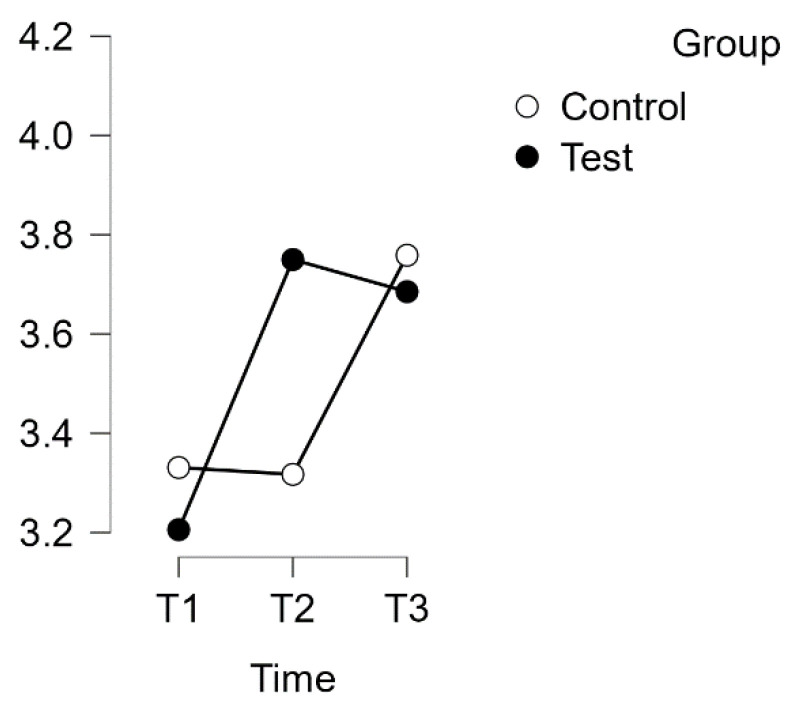
Estimated marginal means for Re-Re subscale: Regeneration Orientation.

**Table 1 healthcare-12-00620-t001:** Age distribution experimental group.

	Age	
	Control Group	Experimental Group
Valid	29	34
Missing	0	0
Mean	49.310	49.088
Std. Deviation	9.111	11.139
Minimum	30.000	27.000
Maximum	67.000	65.000

**Table 2 healthcare-12-00620-t002:** A summarised account of the overall PSQ results across all scales.

Post Hoc Comparisons—Group ✻ Time											
			95% CI for Mean Difference					95% CI for Cohen’s d			
		Mean Difference	Lower	Upper	SE	t	Cohen’s d	Lower	Upper	p_tukey_	
Control, T1	Test, T1	7.360	−7.176	21.896	4.835	1.522	0.385	−0.383	1.152	0.651	
	Control, T2	1.093	−8.012	10.199	3.041	0.360	0.057	−0.421	0.535	0.999	
	Test, T2	−11.462	−25.998	3.074	4.835	−2.371	−0.599	−1.377	0.179	0.177	
	Control, T3	−10.977	−20.082	−1.872	3.041	−3.610	−0.574	−1.076	−0.071	0.006	**
	Test, T3	−5.875	−20.411	8.661	4.835	−1.215	−0.307	−1.072	0.458	0.829	
Test, T1	Control, T2	−6.267	−20.803	8.269	4.835	−1.296	−0.328	−1.093	0.438	0.787	
	Test, T2	−18.822	−27.231	−10.413	2.809	−6.702	−0.984	−1.501	−0.467	<0.001	***
	Control, T3	−18.337	−32.873	−3.801	4.835	−3.792	−0.959	−1.763	−0.155	0.003	**
	Test, T3	−13.235	−21.644	−4.826	2.809	−4.712	−0.692	−1.172	−0.212	<0.001	***
Control, T2	Test, T2	−12.556	−27.092	1.980	4.835	−2.597	−0.656	−1.438	0.125	0.107	
	Control, T3	−12.071	−21.176	−2.966	3.041	−3.969	−0.631	−1.138	−0.124	0.002	**
	Test, T3	−6.968	−21.504	7.568	4.835	−1.441	−0.364	−1.131	0.403	0.702	
Test, T2	Control, T3	0.485	−14.051	15.021	4.835	0.100	0.025	−0.735	0.786	1.000	
	Test, T3	5.587	−2.822	13.996	2.809	1.989	0.292	−0.157	0.741	0.354	
Control, T3	Test, T3	5.102	−9.434	19.638	4.835	1.055	0.267	−0.497	1.031	0.898	

** *p* < 0.01, *** *p* < 0.001. Note. Computation of Cohen’s d based on pooled error. Note. *p*-value and confidence intervals adjusted to compare a family of 15 estimates (confidence intervals corrected using the Bonferroni method).

## Data Availability

The data presented in this study are available from the corresponding author on request but are not publicly available due to data protection for privacy reasons. The Consort 2010 information checklist has been followed and is available on request.

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
