# Peer review of "Evaluation of a Four-Week Online Resilience Training Program for Multiple Sclerosis Patients"

_healthcare, 2024, doi:10.3390/healthcare12060620_

Round 1
Reviewer 1 Report (Previous Reviewer 1)
Comments and Suggestions for Authors
Comments:
This is the 3rd submission by the authors. The paper was rejected by editors after 2nd review with suggestions for substantial improvement. I could not find any substantial improvement in the revised submission.
I could not find my previous comments in the dashboard, (Probably for the technical team of the journal to address) or with the submitted manuscript, so I cannot verify all the points raised by me previously. However, some points, I remember have not been addressed properly. Authors need to address the following points.
1- As this is an interventional study, Information about prior registration is necessary. As per the explanation of authors, this study was not registered a priori, and they have applied for registration after completion of the study. This is not in sync with CONSORT guidelines. Moreover, the authors need to clearly mention this in manuscript, which has not been done in this resubmission.
2- The total sample size in first submission was 44, but now changed to 38. I am not sure how sample size can change during different submissions?
3- The inclusion and exclusion criteria should be clearly mentioned with specific points. It was highlighted in previous comments but has not been mentioned in the manuscript.
4- Table 3, is gender distribution or age?
5- No addressal of important covariates. Factor such as disease severity, disability, depression and other psychiatric interventions can directly affect the outcome of resilience programme. The study design has not addressed these points. The authors had mentioned some justifications in previous replies but those were not incorporated in the manuscript discussion or limitations.
6- The conclusion is written before the discussion part. Should be corrected.
7- Authors should discuss about the limitations of the study.
8- Mentioning the study as “pilot study” in resubmission stage is not a good idea. Consideration of disease stage and co-morbidities can be done in Pilot study also.
Comments on the Quality of English Language
Minor editing of the manuscript is needed/
Author Response
Dear reviewer,
We sincerely appreciate the thoroughness with which you have approached the review process, making insightful suggestions and raising thought-provoking questions that have undoubtedly strengthened the overall coherence and rigour of our research. We have carefully considered each suggestion and made every effort to incorporate all changes and address all concerns raised, and we hope that our changes and improvements meet your expectations.

Reviewer 2 Report (Previous Reviewer 3)
Comments and Suggestions for Authors
I am honored to have been assigned to review your paper. Thank you for providing a clear and understandable explanation regarding my concerns about participant attrition. I understand that the research team sent out three reminders to all participants, which shows a proactive effort to enhance participant engagement. Your decision to maintain the anonymity of the research is an important measure in adhering to research ethics, and I acknowledge that it is inevitably challenging to precisely identify the reasons for participant dropout under such circumstances. Overall, your study makes a significant contribution to evaluating an online resilience training program for patients with multiple sclerosis. I am grateful for your effort and dedication, and I look forward to the positive impact this research will have on the broader community.
Regarding the Hawthorne effect and biases due to non-blinded allocation, this study focuses on the effectiveness of the intervention, participant compliance, and the statistical significance of observed changes without directly addressing the potential influence of these factors on the results. Please add discussion on this.
In the revised manuscript, the order of discussion and conclusion has been changed. Please revise this.
Author Response
Dear Reviewer,
Thank you very much for your review and for signing off on our paper. We are pleased to hear that you found our study to be valuable, and we are especially grateful for your positive comments. Your insights have been invaluable in helping us to refine our research and enhance its contribution to the field.
In response to your suggestions, we have taken great care to elaborate on the limitations of our study in the discussion section. We have thoroughly considered the potential constraints and challenges inherent in our research, and we believe that addressing these limitations enhances the transparency and credibility of our findings. Specifically, we have discussed the Hawthorne effect, potential attrition bias, and the limitations of the study span. By acknowledging these factors and discussing their potential impact on our results, we hope to provide readers with a more comprehensive understanding of the scope and implications of our research.
Also, we have corrected the order of the discussion and conclusion part.
Thank you again for your time and consideration.
Round 2
Reviewer 1 Report (Previous Reviewer 1)
Comments and Suggestions for Authors
The authors have addressed my previous queries adequately and clarifications have been provided regarding sample size and registration. Necessary changes have been made in the manuscript. The trial has been registered at a data base.
This manuscript is a resubmission of an earlier submission. The following is a list of the peer review reports and author responses from that submission.
Round 1
Reviewer 1 Report
Comments and Suggestions for Authors
Comments:
This is a randomized study evaluating the efficacy of a 4-week online resilience training programme in patients with multiple sclerosis. The research question is clinically interesting. There are several concerns about the study design and reporting of results which I have enumerated below. The authors need to address the following points:
1- This is a RCT. Hence CONSORT guidelines should be followed while reporting the results. A statement regarding the same should be included in the manuscript.
2- A CONSORT flow diagram should be provided for better understanding.
3- The explanation of the study in line 92- 98 is confusing. It is mentioned that 30 patients did not complete all 3 questionnaires and hence excluded from the study. And then remaining 64 participants were assigned to two groups. While in the first line it is mentioned 64 participants completed a 28-days resiliency training course (Hence the intervention). Hence, there is no clarity how many participants were screened, how many received the resiliency course.
4- What were the inclusion and exclusion criteria? Was severity of MS taken in to consideration?
5- Was any stratified randomization based on age, gender, disease severity etc considered?
6- Was the allocation blinded?
7- Important covariates, e.g disease severity, neurological disability, co-existent depression, anxiety etc has not been assessed. No data is available to determine if these parameters were balanced between the groups at baseline or not.
8- As this is an interventional study, was it registered at any registry? If so, the registration number should be provided.
9- How the assessment done? Was it blinded? If the questions were self-administered, then the participants filled during scheduled visit after 4 weeks or via any other medium?
10- How the compliance for the training programme was ensured?
11- Authors mention about 30 patients being excluded from study due to non-completion of three questionnaires? Which one was the third questionnaire?
12- Was Intention to treat analysis done?
13- The authors need to brief about the scales mentioning the score range and its interpretation.
14- Baseline clinical parameters for each group should be provided in a tabular form.
15- What the Figure -6 represents? It is PSQ sub scale or Re-Re scale?
16- In the statistical analysis part, the authors mention about evaluating the internal consistency of the different scales. The results and the Cronbach alpha should be mentioned under the results section.
Comments on the Quality of English LanguageAuthor Response
Dear Reviewer,
I would like to express my sincere gratitude for your thoughtful review of our paper. We have addressed all of your questions and comments in the attached document, and corresponding changes have been made in the main manuscript. Your input has been immensely valuable, and we truly appreciate your time and expertise.
Thank you once again.

Reviewer 2 Report
Comments and Suggestions for Authors
The design is a controlled randomized 4-weeks trial during which 34 PwMS listened daily to videos in 2 sessions (AM and PM) designed to improve resilience. They were asked to complete questionnaires measuring stress related issues, at onset, after 4 weeks, and 3 months later.
The 30 controls just answered the 3 questionnaires.
Please consider the following questions and commentaries.
· There are no details other than age and gender on the MS course: disease duration, severity, treatments, relapse frequency and severity, education level, life habits (tobacco, alcohol, and substance use, leisure activities, exercise), comorbidities, etc.
· Are there differences on these parameters between these 2 groups?
· Was there ethnic diversity among participants?
· Did you measure the level of observance for the sessions? It would be surprising if all the members of the trial group attended all the 56 sessions.
· You should describe the 30 PwMS who did not complete the study.
· Provide details, or references, on the videos selected. Were they tested in other populations?
· The figures provided are very similar, showing a decrease of the improvement over time.
· In the same way, the controls show an improvement, often significant, between T2 and T3. How do you explain this?
· Did some of the PwMS experience a post traumatic stress syndrome?
· This is a short and intense trial. Provide suggestions on how this could be implemented in regular practice.
Author Response
Dear Reviewer,
I would like to express my sincere gratitude for your thoughtful review of our paper. We have addressed all of your questions and comments in the attached document, and corresponding changes have been made in the main manuscript. Your input has been immensely valuable, and we truly appreciate your time and expertise.
Thank you once again.

Reviewer 3 Report
Comments and Suggestions for Authors
It's an interesting study, but overall it needs a lot of revision.
1. Introduction
In the paragraph below 'In summary, ~', please add information about research on stress in MS patients.
Also, please add prior research on online self-directed training to the paragraph above the purpose.
2. Materials and Methods
The analysis method was used to calculate the sample size in the GPower program. Since there were three-time points in this study, repeated measured ANOVA must be used. Please re-calculate the sample size.
The regional associations of the DMSG received a description of the study by email from the researcher. Please explain how they advertised the study to their members. Additionally, interested research subjects are requested to describe in more detail how they expressed their intention to participate in the study to the researcher.
Please describe the data collection period t1, t2, and t3 in more detail.
Was the 4-week online self-directed training conducted between t1 and t2? Please describe in detail when it took place.
Please provide detailed operating protocols for online self-directed training so that other researchers can conduct repeated studies.
How many neurology patients at the Bundeswehrkrankenhaus Hamburg and the DMSG each agreed to participate in the study?
Did those who said they would participate take the pre-survey (t1) right away?
Was random assignment done after a preliminary survey?
When exactly was randomization done?
Were there any considerations for the subjects before random assignment?
Please clearly indicate the inclusion and exclusion criteria for research subjects.
What online randomization tool was used in this study? Please explain in more detail.
Please present the CONSORT flow diagram as a figure.
Why was the number of samples for the two groups different during randomization?
Was this study blinded (subjects, data collectors, program operators)? Please describe in detail how you handled bias that affected the research results.
In the two research tools, please add the concept, number of sub-domain items, explanation of the measurement scale, and interpretation of the scores.
Please add criteria for removing outliers in the data analysis method.
3. Results
There were 30 people in the control group, but the results show 29 people. Please check.
Why were the test group's total PSQ score and sub-domain scores higher at t2 than at t1?
Please check if the title of 'Figure 6. Estimated marginal means for PSQ subscale: Worries.' is correct.
4. Discussion and Conclusion
Please have an in-depth discussion based on the research results.
Author Response

(The authors gave the same response as above.)

Round 2
Reviewer 1 Report
Comments and Suggestions for Authors
Comments:
The authors have responded my previous comments. However, many points remain inadequately answered. The manuscript has not been improved significantly. Moreover, some changes are done after the initial comments. This suggests that the RCT was not conducted in a pre-specified way. I have following comments to make:
1- This is an interventional trial, however, the authors have acknowledged that it was not registered prior and the authors are planning to register it now (after completion of the study). I think this does not adhere to the CONSORT guidelines for RCT.
2- The explanation of the authors (query 4), suggest that no defined inclusion criteria was considered. The explanation given in line 99-102, suggests only interested participants. No information about age cut off , severity of MS (EDSS cut off). (authors have mentioned they have not assessed the baseline severity) , other co- morbidities like psychiatric illness. People who are disables enough to carry out the interventions, should have been ideally not included in the study. This is a major concern with the study design which is adversely affecting the internal and external validity.
3- In response to query 7, the authors mention that they decided against the collection of clinical information and decided to focus on psychological interventions. But in a RCT baseline similarities are essential to reach a definite conclusion. In this study, neither the clinical information, nor the baseline psychological elements e.g depression, anxiety etc has been provided. They are important co-variates and influence the outcomes directly. Non-addressal of them is a major flaw in the study design.
4- There are about 32% loss to follow up. The authors have not conducted any Intention to Treat analysis. Large loss to follow up diminishes the power of the study. Interestingly, the authors mention (query 12), that they did not had any drop outs.
5- In the revised manuscript, the conclusion part starts with finding of the study. There is no coherence between the discussion part and conclusion.
6- The abstract, mentions 64 patients were recruited in the study, while the CONSORT flow diagram mentions 94 were randomized.
Comments on the Quality of English Language
Minor editing of the English language is required.
Reviewer 2 Report
Comments and Suggestions for Authors
Thank you for carefully addressing the reviewers' and suggestions.
A shorter more concise text would improve readability.
Author Response
Thank you for your kind reply
Reviewer 3 Report
Comments and Suggestions for Authors
Although it is very difficult to verify experimental effects in clinical trials with patients and report long-term effects through follow-up investigations, I believe researchers should continue to conduct such experiments and report the results.
The authors faithfully revised the reviewer's comments in a short time. However, there are still some areas that require further revision.
There must be an appropriate reason for the 30 people dropping out during the three measurement periods. What steps did the researchers take to encourage those who did not respond at T3 to respond? Please describe the reason for attrition in detail.
In the tool description, it would be easier for readers to understand that 'Joy', a subdomain of the PSQ, means 'lack of joy'.
Even if online training was provided to the experimental group, there may have been a Hawthorne effect in the experimental group due to a lack of blinded allocation because both groups were briefed about the study. Authors should describe the bias in reporting results that may occur due to these errors as a limitation.
Why is the social term ‘gender’ used in this study? Unless there is a special reason, it would be better to use the biological term 'sex'.
I understand that trial registration is usually done at the experiment planning stage, but it is difficult to understand that it is being registered. Please provide a valid reason for the delay in registration.
